# Analysing the sensitivity of a flood risk assessment model towards its input data

Hanne Glas[1], Greet Deruyter[1], Philippe De Maeyer[2], Arpita Mandal[3], Sherene James-Williamson[3]

[1]Department of Civil Engineering, Ghent University, Ghent, 9000, Belgium
[2] Department of Geography, Ghent University, Ghent, 9000, Belgium
[3]Department of Geography and Geology, University of the West Indies, Mona Campus, Jamaica

*Correspondence to*: Hanne Glas (hanne.glas@ugent.be)

**Abstract.** The Small Island Developing States are characterized by an unstable economy and low-lying, densely populated cities, resulting in a high vulnerability to natural hazards. Flooding affects more people than any other hazard. To limit the
consequences of these hazards, adequate risk assessments are indispensable. Satisfactory input data for these assessments is hard to acquire, especially in developing countries. Therefore, in this study, a methodology was developed and evaluated to test the sensitivity of a flood model towards its input data in order to determine a minimum set of indispensable data. In a first step,  a flood damage assessment model was created for the case study of Annotto Bay, Jamaica. This model generates a damage map for the region based on the flood extent map of the 2001 inundations caused by Tropical Storm Michelle. Three
damages were taken into account: building, road and crop damage. Twelve scenarios were generated, each with a different combination of input data, testing one of the three damage calculations for its sensitivity. One main conclusion was that population density, in combination with an average number of people per household, is a good parameter in determining the building damage when exact building locations are unknown. Furthermore, the importance of roads for an accurate visual result was demonstrated.

**Keywords:** *SIDS, flooding, risk assessment, damage map, sensitivity analysis*

## 1 Introduction

Natural hazards have a great economic impact on countries worldwide. The losses as a result of earthquakes, cyclones, landslides, flooding and tsunamis are estimated at up to 300 billion USD per year (UNISDR, 2015). Natural hazards do not only cause economic but also human losses. Between 1975 and 2008, over 2.2 million people died due to natural hazards
worldwide (ISDR, 2009).  Floods affect more people worldwide than any other hazard (UNISDR, 2015). Not only low-income countries suffer from severe inundations.

Low-lying, densely populated areas with unstable economies have little protection against natural hazards (UNESCO, 2014). Many of these areas can be found in the SIDS (Small Island Developing States), which are located in the regions of Latin America, the Caribbean, East Asia and the Pacific, and are expected to lose 20 times more of their capital stock in disasters

each year than Europe and Central Asia (UNISDR, 2015). In Jamaica, for example, economic damage due to flooding was estimated at 1.5 billion USD over a period of four years (ODPEM, 2013b).

To limit the consequences of flooding, many governments revert to technical interventions, such as dams, levees and flood forecasting. These approaches, however, have shown limited success in several countries (Gall et al., 2011; Deckers et al., 2010), leading to new approaches that focus on flood risk management rather than flood control (Institute for Water Resources, 2009). One of these approaches is a quantitative flood risk assessment, indicating the high-risk areas by estimating the possible damage caused by a flood hazard. The output of this method can help decision makers in identifying the most vulnerable regions and allocating the right resources and funds to the right locations. The technocratic interventions, as mentioned before, can thus be applied more effectively and sensibly.

In many developed countries, a risk-based flood tool has been developed, for example the HIS-SSM model for the Netherlands (Kok et al., 2005), the LATIS model for Flanders, Belgium (Vanneuville et al., 2005), the HAZUS-MH Flood Model for the USA (FEMA, 2009) and the FLEMO model for Germany (Apel et al., 2009). The use of such risk assessment models has, however, been limited, due to questions about the uncertainty and reliability of the results (Merz et al., 2004). Since these methodologies are built on input data that each have their own accuracy and uncertainty, the output of the methodology has an uncertainty that is very difficult to quantify (Yu et al., 2013). Furthermore, an increase of the input data accuracy doesn't automatically imply a decrease of the output's uncertainty (Apel et al., 2009). Nonetheless, the existing models are being optimized and are used as decision tool in urban planning projects as is the case in Flanders, Belgium (Deckers et al., 2010).

In developing countries, the limited data availability forces researchers to find other types of input data for flood damage and risk assessments. Kumar and Acharya (2016), for example, have performed a flood risk assessment in Kashmir Valley, India, using satellite imagery as input. Kwak et al (2015) created a rice crop damage map for the Cambodian floodplain using satellite imagery combined with a DEM and land use data. Other studies have attempted to provide adequate damage and risk results by using vector data, for example the risk assessment for Annotto Bay, performed by ODPEM (2013a).

Since the necessary input data is hard to find in developing countries, a thorough assessment of the data needed should be done. What are the minimum data requirements to build a reliable model? What is the sensibility of the model to the different datasets? These are the questions that need to be answered whilst keeping in mind that a certain degree of uncertainty is inherent to the methodology.

This paper investigates the different types of data used in a flood risk assessment for Annotto Bay, Jamaica, and their influence on the overall result by performing a sensitivity analysis on the risk assessment model with different combinations of input data. The output of every combination is tested on its accuracy based on the estimated total material loss and affected area and the geographic positions of high- and low-risk areas, compared to the benchmark output that uses all available data.

## 1.1 Sensitivity analysis

Data and methodology uncertainties are inherent to every risk assessment model (Carrington & Bolger, 1998). Since they can influence decision-making, these uncertainties have been quantified in several previous studies (Yu et al., 2013; Apel et al., 2004; Apel et al., 2008; Weichel et al., 2007). More and more exact data, however, does not always translate in a
decrease of the uncertainty, since the influence on the final result differs for each input data set (Apel et al., 2008).

In many SIDS, geographic and statistical data availability is a major issue. Moreover, the data available has a questionable accuracy (Glas et al., 2015). It is therefore important to define the importance and influence of every input data set. With a sensitivity analysis, the influence of all input data on the overall result and its degree of detail is determined. When the sensitivity of a model towards its input is known, the minimum required data and the level of detail in order to get an
accurate result, can be deduced. Although uncertainty analyses are frequently performed in the literature, sensitivity analyses to determine the necessity of the input data are rare. Nonetheless, this information is useful in setting up an uncertainty analysis. The impact of an input data set on the final result can serve as an indication of the impact of the uncertainty of this data set on the overall result and its uncertainty.

In this study, the input of a flood risk assessment performed for Annotto Bay, Jamaica (Glas et al., 2015), was used as case
study for the sensitivity analysis, because in 2012 a lot of accurate data was collected for this town in the framework of another research program (ODPEM, 2013a). Since hydraulic and rainfall data is scarce in this region, and return periods of floods are unknown, this quantitative risk assessment focuses on material damage due to inundations caused by the Tropical Storm Michelle, in 2001 (WRA, 2002).

## 1.2 Study area

Annotto Bay is a small coastal town in the northeast of Jamaica. The town is vulnerable to several natural hazards, of which storm surges and riverine flooding are the most severe (ODPEM, 2013a). This is due to the high-risk location of the community. Not only is the town situated close to the coastline, but it is also enclosed by the Blue Mountains. This topography, together with the presence of four rivers traversing Annotto Bay, causes the rapid flooding of the community whenever perpetuation occurs in the mountains (WRA, 2002). Since the highest point of the town is only three meters above
Mean Sea Level, Annotto Bay suffers severely from storm surges as well. There are about 5,500 inhabitants in the area, living mainly in concrete and wooden buildings (Statistical Institute of Jamaica, 2012). The land use in the study area and the locations of the rivers, roads and buildings is shown in Figure 1.

All damage calculations made in this study were based on the flood map of the inundations on both the 28[th] and the 29[th] of October, 2001, caused by Tropical Storm Michelle.  The city of Annotto Bay was largely flooded for two days (Figure 2).
Houses, infrastructure and crops were damaged, however, since the flow velocity was less than 0.3 m/s, there was only little severe structural damage (ODPEM, 2013a).

## 2. Methods and results

In this chapter, the methods and results of the sensitivity analysis are discussed. In a first step, a benchmark flood risk model was determined. This model was created using all available data and was based on the Flemish LATIS methodology (Deckers et al., 2010) and on a risk assessment performed by ODPEM (2013a). In the benchmark risk assessment, geographic information was combined with the replacement values of the elements at risk and with the damage factors. Replacement values represent the cost to rebuild an element when it is totally destroyed, while the damage factors are an estimate of the degree of destruction based on the flood level, in feet, at the location of the element at risk. Hence, the damage factor will be a number between 0 and 1, with 0 being no damage at all and 1 being complete destruction. The three types of elements at risk that suffered most damage according to ODPEM (2013a) were buildings, crops and roads. Due to limited information on other types and the impact of the flooding on these elements at risk, only the damage costs for buildings, crops and roads were calculated by multiplication of the replacement value by the damage factor to generate a damage map, indicating the total damage cost per square meter for the study area. The input data of this model is listed in Table 1.

This first assessment, the benchmark, is called Scenario 1 (S1). Eleven other scenarios, each with less, or less detailed, input data than S1, were tested and compared to this first one. Table 2 shows an overview of all scenarios and Table 3 provides a matrix showing what data was used in which scenario. The scenarios are discussed per sensitivity. Four types were tested: building damage sensitivity, road damage sensitivity, crops damage sensitivity and data type sensitivity. In each section, the methods are discussed first, followed by the results.

For each scenario, four elements were compared: the spatial difference, the visual output, the total damage cost and the total damaged area. To test the first element, all damage maps were converted into raster maps with a resolution of 5 meters. Then, the value of every pixel was compared to the values of its neighbors. The spatial difference is defined in Eq. (1) as the probability that a pixel has a different value than its neighbor:

$$SD = \frac{\sum_1^n \frac{P_{sd}}{P_s}}{n} \tag{1}$$

where SD is the spatial difference, $P_s$ the number of neighboring pixels, $P_{sd}$ the number of neighboring pixels with a different value and n the total number of pixels. The concept of spatial difference is also demonstrated in Figure 3. The value of the spatial difference is thus a tool to describe the level of detail of a damage map. Since the resulting damages were assigned to classes in the final maps, this level of detail would be difficult to deduct from only the visual mode of representation. Together with the total damage cost, which is the sum of the calculated building, road and crops damages, and the total damaged area, the visual result and the spatial difference determine the influence of each type of data on the overall result.

All scenarios were modeled in ArcGIS 10.2 using Python. The methodology of the risk assessment was automated through a script written in the ArcPy module. Although small differences exist between the scenarios, caused by the use of different or less input data, the overall methodology remains the same.

## 2.1 Benchmark map

### 2.1.1 Method

To generate the benchmark map, three types of damages were assessed. Building damage calculations were based on the exact GPS position of all of the buildings in Annotto Bay, as well as their building materials and the number of floors (ODPEM, 2013a). By using average Jamaican market values, calculated by ODPEM (2013a) for the material cost and the building surface area, a maximum damage value was determined per building. Subsequently, the real damages were calculated by multiplying these maximum damage values with a damage factor based on the water levels. The damage factor were transferred from Japanese damage functions, as retrieved from Dutta et al (2003), and the water levels were retrieved from the 2001 flood map (ODPEM, 2001). The Japanese damage functions could be transferred to Jamaica due to the similarities in geography and building engineering procedures. Most Japanese and Jamaican buildings are constructed in a similar manner with solid concrete or wooden walls. The distinction between these two building types is made in the damage functions as well as in the building database of Annotto Bay. The calculated real damages were then summed up per land use polygon, in order to generate a clear view of the building damage.

The damage to roads was calculated using the road network dataset (ODPEM, 2013a). This dataset divides the roads into four classes, each with their own properties, for example the width of the road. The line dataset was converted into polygons, based on the different widths. Using an average maximum road damage, calculated by Collier et al (2013) for developing countries, and combining this with damage factors from the Flemish LATIS flood risk assessment tool (Deckers et al., 2010), the real damage was then calculated for all roads.

Finally, the crop damage map was generated. A difference was made between banana plantains and other crops, due to the different reaction to inundations and the different average cost of the crops. As banana plants can only survive water saturated conditions up to 48 hours because of their fragile roots (Rajamannan, 2004), the duration of the flood is especially important for these plants, since a two-day flood, as this was the case in 2001, causes 100 percent destruction of the plants. For the damage calculations of the other crops, an average was used of the damage factors of eight crop types defined by Dutta et al (2003). These crops are commonly cultivated in Japan as well as in Jamaica. Therefore, the crop damage functions could also be transferred. The maximum crop damage value was based on information from FAOSTAT (2014) and was multiplied with this damage factor to determine the crop damage cost. Since the damage factor for the banana plantains was 1, their real damage value was equal to the maximum damage value.

Since there is only very limited information on the exact consequences of the 2001 flood, the benchmark model could not be validated. However, the small amount of information that was available, could serve as an indication. The number of affected houses, for example, was 749 (ODPEM, 2013a), while the benchmark model calculated this at 799. The overestimation can be explained by the generalization done by the model, that does not take into account the fact that some houses will resist better than others and will thus have no damage. There was no comparable data for road and crop damage.

The lack of validation increased the uncertainty of the model. However, this research did not take into account the uncertainties of the input data or the model, since the aim of this research was to investigate the sensitivity of the model towards its input data. Hence, to identify the influence of each type of input data, S1 was an acceptable benchmark.

### 2.1.2 Results

The benchmark damage map visualizes the output of the flood risk assessment model for Annotto Bay, as shown in Figure 4. Table 4 contains the three numeric elements on which the comparison of the scenarios is based: the total damage, the total damaged area and the spatial difference, as calculated for S1. The total damage cost is calculated at 7.49 million USD, of which 7.08 million USD, or 94.6% is damage to buildings.

## 2.2 Building damage sensitivity

### 2.2.1 Methods

In the next four scenarios, the sensitivity of the flood risk model towards the data used to calculate building damage was investigated. In S2, the information concerning materials and the number of floors was removed and replaced by average values for all buildings in Annotto Bay. In S3, the location of the buildings was also eliminated, leaving only the number of

15 buildings in the total study area as information. In this scenario, after testing the available data in and around the study area, including the exact building locations and the land use data, 90% of the buildings was presumed to be in urban areas and the other 10% in rural areas. In S4 and S5, population information was used to determine the building damage, based on the average number of 3 people per  building (WRA, 2002). In S4, the population density per statistical sector was used to calculate the number of buildings. In S5, however, only the total number of people in the study area was known. Here, the

20 same assumption was made as in S3 about the division of buildings between rural and urban areas.

### 2.2.2 Results

Figure 5 shows the visual result of the four scenarios, while Table 5 shows the calculated damage, the damaged area and the spatial difference in comparison to the benchmark results of S1. Visually, no big changes can be observed in the indication of the high-risk areas. The slightly lower spatial difference in S3 and S5 does indicate a decrease in the level of detail. While

S2 gives the result that is most similar to the result of S1, the table clearly shows an important difference of 19.75% in the calculation of the total damage cost. This percentage rises to 20.88% when only taking into account the building damage. Although the visual result of S4 is less detailed than the benchmark, the spatial difference of 0.045 indicates a similar level of detail as in S1. Moreover, this scenario gives the best result towards the calculation of the total damage. The calculated building damage of S4 is 6.59 million USD, which is 6.96% lower than the calculated building damage in S1.

### 2.3 Road damage sensitivity

#### 2.3.1 Methods

Scenarios 6, 7, 8 and 9 were used to assess the sensitivity of the risk assessment towards the road data. In S6, the road classes were presumed to be unknown, giving all roads the same average width. S7 did not take the roads into account. In S8, the location of the roads was eliminated and therefore, they were calculated as a percentage of the land use. After analysing the available data in and around the study area, the percentages were set at 5% roads in urban areas and 2% in rural areas. S9 only used the road network to divide the land use polygons, but did not take them into account in the damage calculations.

#### 2.3.2 Results

The road cost is only a small share of the total calculated damage. This is clear when comparing the total damage of the four scenarios to the damage of the benchmark in Table 6. S6, for example, generates almost identical numbers as S1. Visually, these scenarios are almost identical. However, when assessing only the road damage, S6 generates a damage cost of 41 thousand USD, which is 20.59% higher than the calculated damage cost of 34 thousand USD in S1.

There is a significant difference in damaged area between S1 and S8. Since the threshold value for road damage is 0 feet and the road damage is spread over the entire study area in S8, all flooded areas have damage. Moreover, visually, S8 shows a different, less accurate, result than the other scenarios, as shown in Figure 6. The scenario has a low spatial difference of 0.018. The total road damage cost of 32 thousand USD, however, is only 5.88% lower than the damage cost in S1.

Although S7 clearly has a better visual result than S8, indicating the areas without any damage more accurately, the spatial difference of this scenario is lower. Due to a larger damaged area in S8, more pixels are taken into account in the spatial difference calculations, increasing the possibility of having neighboring pixels with a different value. The level of detail is thus higher in S8, but the visual result shows large deviations from S1. The removal of the roads in S7 and S9 only has a small effect on the total damage and damaged area, but it does have an important influence on the level of detail, as proven by the spatial differences. The ninth scenario, nonetheless, does have a more accurate visual result then the other road scenarios, due to the use of the road network to divide the land use polygons.

### 2.4 Crops damage sensitivity

#### 2.4.1 Methods

S10 tested the sensitivity of the model by assuming the difference between banana plantains and other crops was unknown. An average maximum damage value was calculated from the values for banana plants and other crops, grown in Jamaica. The damage factor used was also an average, but only of the damage factors of other crops, since the damage factor for banana plants was 100% for every water depth, due to the duration of the flood.

#### 2.4.2 Results

Since the real damage value of the crops is rather small in comparison to building damage values, S10 only has a small effect on the result. Therefore, the visual view of the map is almost identical to the benchmark damage map. This can be seen in Figure 7. Furthermore, Table 7 demonstrates that the calculated total damage and damaged area differ only little from the values that were generated by the model used for S1. However, the crop damage cost of 154 thousand USD in S10 is 58.60% lower than the crop damage cost of 372 thousand USD in S1.

## 2.5 Data type sensitivity

### 2.5.1 Methods

The last two scenarios looked into the sensitivity of the model towards the input data type. In the benchmark model, all input data was vector data. In areas with little data available, however, a lot of information will have to be gathered from satellite imagery. Therefore, all input data in S11 and S12 was converted to raster data with a resolution of 10mx10m for S11 and 30mx30m for S12 to simulate satellite data. The former resolution was chosen since several commercial high-resolution satellite systems, e.g. SPOT, provide images with a world coverage with this resolution. The Landsat program uses the latter resolution and provides free images through an online service. The calculations for the building damage were based on population data, in the same way as in S4.

### 2.5.2 Results

Although the two damage maps, as shown in Figure 8, visually do not differ a lot from the maps of S1 and S4, Table 8 shows that the total damage cost is substantially higher than the cost in S1 and S4. All three separate damage costs show a large overestimation compared to S1 and S4. The road damage cost, especially, is 27 times larger in S11 and even 78 times larger in S12 than in S1. This is due to the fact that road damage is calculated per pixel, and the pixels in both scenarios have a resolution larger than the width of the roads. Hence, the area assigned to roads is overestimated.

The total damaged area is also slightly larger, due to the conversion of the polygon flood map to a raster map. Since the input of the scenarios was raster data, every pixel has been calculated separately. Therefore, the level of detail, and thus the spatial difference, is higher than in S7, S8 and S9. When comparing the results of S11 and S12, it can be stated that the spatial difference shows a growing decrease of accuracy as the resolution of the raster data increases. Moreover, the visual result is less detailed and gaps arise in the final map.

## 3. Discussion

In all scenarios, more than 90% of the total flood damage consists of building damages. Consequently, scenarios that test the models sensibility for building data show the largest deviations in the total damage. Figure 9 shows the deviation for every scenario from the total cost of S1.

When looking at the scenarios focussing on building damage, S4 has the best result, with a deviation of 6.58% in relation to the result of S1. This scenario has calculated the damage cost based on population density per statistical sector. In the case study of Annotto Bay, the benchmark study made use of the exact GPS locations of all buildings in the region. In many other areas in the SIDS, this detailed information is not available. Population data, however, exists for most regions free of charge.

Since the model gives a good result, visually as well as in the total damage cost, this scenario must definitely be investigated further. The importance of an accurate average number of people per household was proven by running the same model with an average of 2 and an average of 4 people per household instead of the average of 3, as given by WRA (2002). When testing the former, the total damage cost of 4.83 million USD is 35.55% lower than S1, while the latter gives a resulting cost that is 21.75% higher than the resulting damage cost of S1.

When only relying on Figure 9, it could be stated that the model is not sensitive to road data at all. However, not only the total damage must be taken into account, but also the spatial impact and the total damaged area have to be included. In Figure 10, the last factor is given. It is clear that S8, the scenario where roads are taken into account as a percentage of the land use, is not a good simplification. Since buildings have a threshold value to be marked as 'inundated' of 1,5 feet, but roads are marked immediately as flooded, the total damaged area in S8 is a big overestimation of the reality. This is affirmed

by the visual result, showing a lot of damaged area with a low cost per square meter.

Although S7 scores very well for the total damage as well as for the total damaged area, the result is a lot less accurate than the benchmark map. This becomes clear when looking at Figure 11, that visualizes the deviation of the spatial difference of all scenarios in relation to S1. In this figure, three scenarios that test the influence of road data have the highest deviation and thus show significantly less detail in their damage map. Although the roads are negligible for the total damage and the

20 damaged area, they are, nonetheless, an indispensable part in creating a visually accurate map.

Visually, as well as in total damage and damaged area, the difference between crops and banana plantains has a small effect on the results, as shown in Figure 11. It must be stated that this is the case for this case study of Annotto Bay, where building damage is the major type of damage. When looking into other regions, where agriculture has a more important role, the difference between crops can be a lot more significant for the results. This has to be further investigated.

Finally, S11 and S12 have shown the sensitivity of the flood model towards the input data type. In this case, all input data was converted to raster data. Although the visual result was similar to the benchmark, there was a clear difference in the total damage and the damaged area. Therefore, vector data has the preference when working in a relatively small study area. When some input data is vector and other raster data, it should be considered to vectorise the last type in order to avoid losing detail. This methodology will give the most accurate result.

**4. Conclusion**

In industrialized countries, several risk-based flood tools were developed to predict and estimate the damages caused by inundations. Although, a lot of detailed data is fed as input for these models, a certain degree of uncertainty is inherent and

can never be fully eliminated. However, such tools are constantly being optimized and are adopted for urban and rural planning in order to prevent damages from future inundations caused for instance by climate change or high degrees of urbanization.

In developing countries the detailed data needed by these models is not available. Therefore, to determine if the methodology used in the developed countries can be transferred to developing countries, it is necessary know what the sensibility of the models is towards the input data.

For this research, a risk-based model inspired by the Flemish LATIS was used for the case study of Annotto Bay. The results show that it is indeed possible to reduce the level of detail substantially, without adding significantly to the model uncertainties.

Since the 2001 flood especially hit the urban areas of Annotto Bay, the building data was the most significant type of data in this study. The scenario that uses the population density and the average number of people per household to calculate the number of buildings as a simplification for the exact location of the buildings produced the best results. The deviation of the total damage cost was only 7% in comparison to the benchmark. As the population data is globally availability, in many cases for free, this is an important finding that can be transferred for case studies in other areas. It must be stated, however, that an accurate number of people per household is indispensable in this scenario.

Another finding of this study is the importance of road data. Although roads have a small effect on the overall cost, they do have a role in the visual end result. An accurate road dataset helps to divide the land use, and to determine the building damage more precisely. In this light, the possibility of using remote sensing images to create road datasets must be investigated, since many available datasets do not include all roads. When using satellite imagery, the road classes cannot be taken into account, but this has been proven to have little impact on the result. Furthermore, a complete dataset can definitely help in defining building damage, since every building must have access to a road and will thus most likely be located close to this road. Combining this information with population data should be investigated further.

No conclusions could be made from the sensitivity analysis towards crop data, because, in this case study, the impact was too small. The results showed little difference between the benchmark scenario, where crops and banana plantains were treated separately, and the scenario where an average cost was used. To further investigate the impact of crop data, a more rural area should be investigated. However, it can already be concluded that the difference between crops and banana plantains can be eliminated in study areas where  urban areas are most affected by flooding.

Finally, the data type plays an important role in the accuracy of the final result of a risk assessment. Using raster data, from satellite imagery for example, causes an overestimation of the total damage and the damaged area, due to the resolution, which causes loss of information detail. Therefore, satellite imagery should always be vectorised before using it as input data in the risk methodology. In further research, more types of raster data with different resolutions should be tested, as well as combinations of raster and vector data.

This sensitivity analysis of the Annotto Bay flood model is a first and important step in determining which data is indispensable and which data can be adapted, replaced or ignored in a risk assessment. Although the road damage has a

small impact on the overall damage cost, this data type is indispensable for an accurate visual result. Furthermore, it is shown that population density data, in combination with an average number of people in a household, is an adequate replacement of the exact housing locations as input data for building damage. Nonetheless, more research should be done in other regions to validate the results of the sensitivity analysis and to investigate the impact to the damage types in different
situations.

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

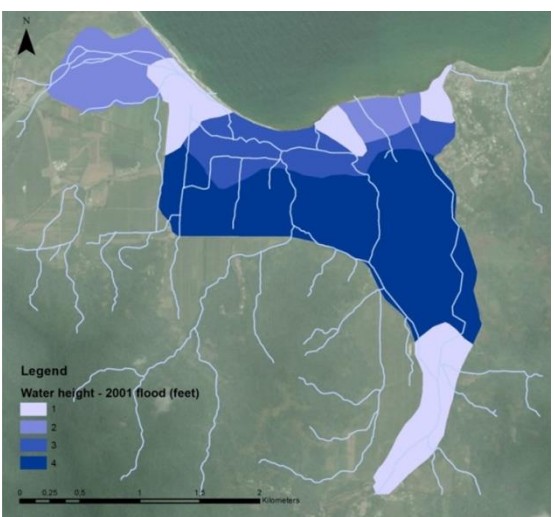

**Figure 2: Flood extent of 2001 inundations caused by Tropical Storm Michelle in Annotto Bay, Jamaica (Glas et al, 2015)**

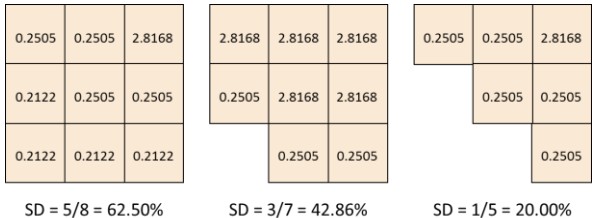

| 0.2505 | 0.2505 | 2.8168 |
|--------|--------|--------|
| 0.2122 | 0.2505 | 0.2505 |
| 0.2122 | 0.2122 | 0.2122 |

SD = 5/8 = 62.50%

| 2.8168 | 2.8168 | 2.8168 |
|--------|--------|--------|
| 0.2505 | 2.8168 | 2.8168 |
|        | 0.2505 | 0.2505 |

SD = 3/7 = 42.86%

| 0.2505 | 0.2505 | 2.8168 |
|--------|--------|--------|
|        | 0.2505 | 0.2505 |
|        |        | 0.2505 |

SD = 1/5 = 20.00%

**Figure 3: Calculation of the spatial difference (SD) of three center pixels with SD = {number of neighboring pixels with different value} / {number of neighboring pixels}**

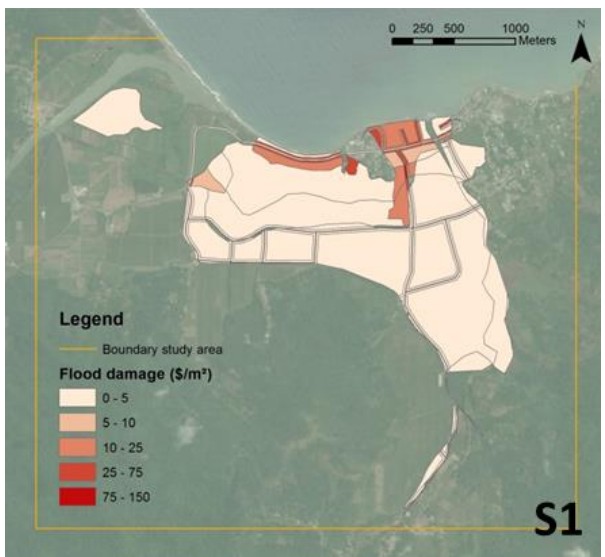

5 **Figure 4: Scenario 1 (S1): Benchmark damage map of Annotto Bay, using all available input data**

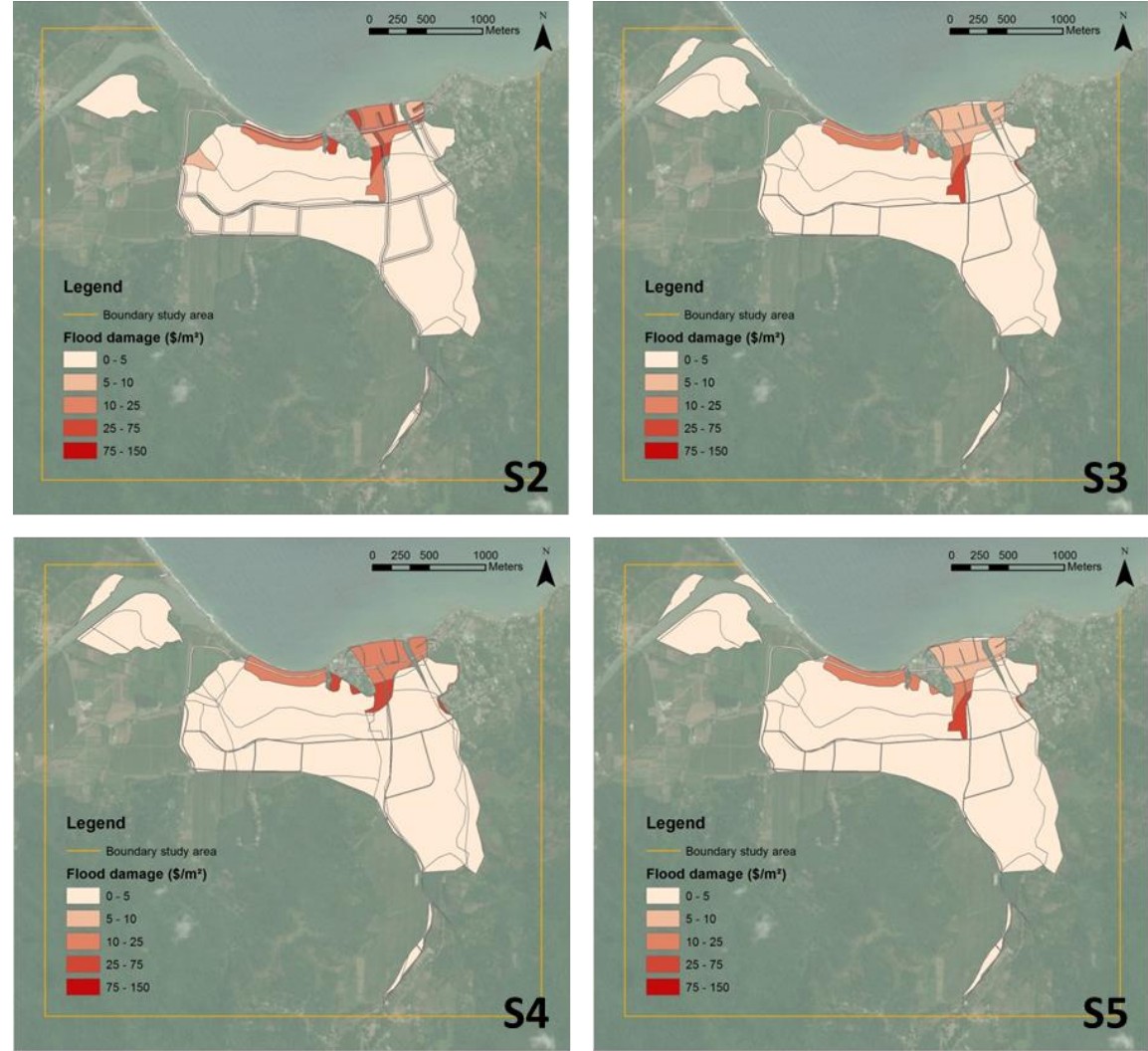

**Figure 5: Damage maps for Annotto Bay for S2, S3, S4 and S5. (Top left: (S2) Building materials and number of floors unknown, Top right: (S3) Building locations, materials and number of floors unknown, Bottom left: (S4) Building density is calculated based on population density, Bottom right: (S5) Building density is calculated based on number of people in study area.)**

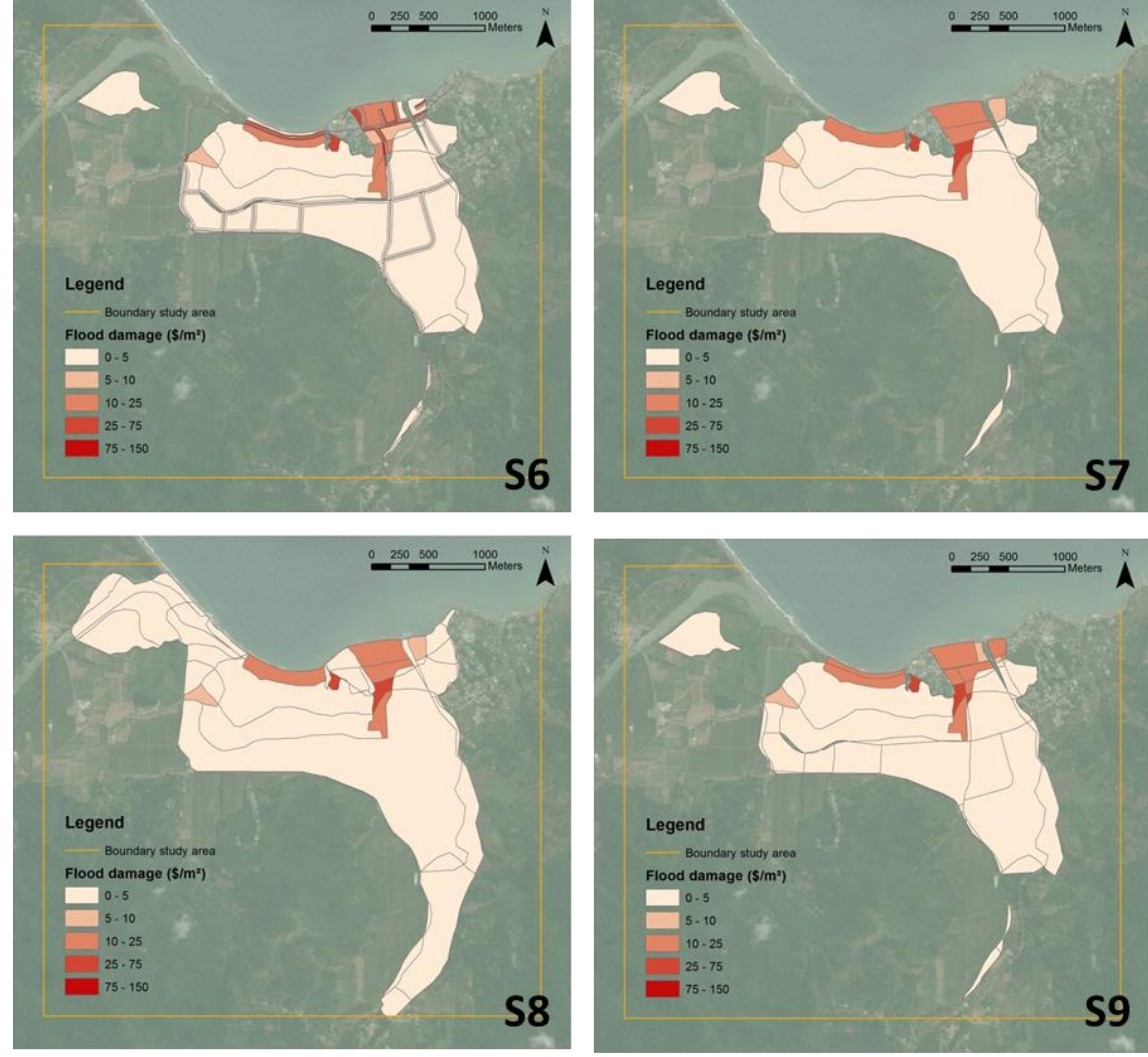

**Figure 6: Damage maps for Annotto Bay for S6, S7, S8 and S9. (Top left: (S6) Road classes are unknown, Top right: (S7) All roads are unknown and not taken into account, Bottom left: (S8) All roads are unknown but taken into account as a percentage of land use, Bottom right: (S9) Roads are only used to divide land use polygons – no road damage.)**

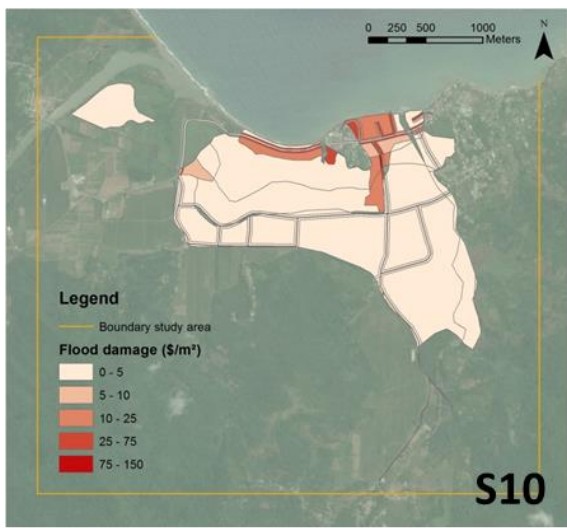

**Figure 7: Damage map for Annotto Bay for S10 (Difference between banana plantains and other crops is unknown.)**

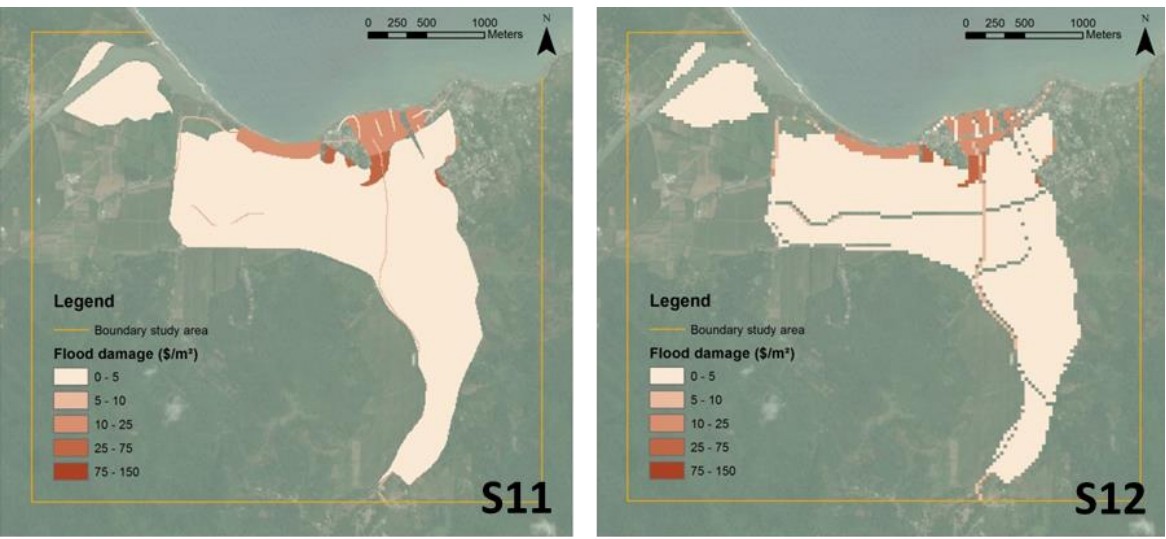

**Figure 8: Damage maps for Annotto Bay for S11 and S12. (Left: (S11) Raster approach (10x10) based on population density, Right: (S12) Raster approach (30x30) based on population density.)**

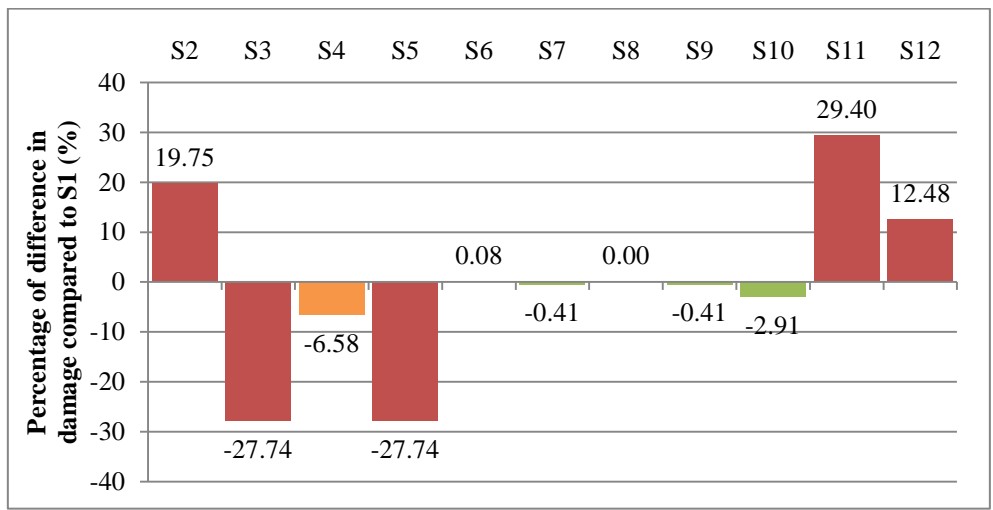

**Figure 9: Deviation of total damage of all scenarios in relation to S1 (=0)**

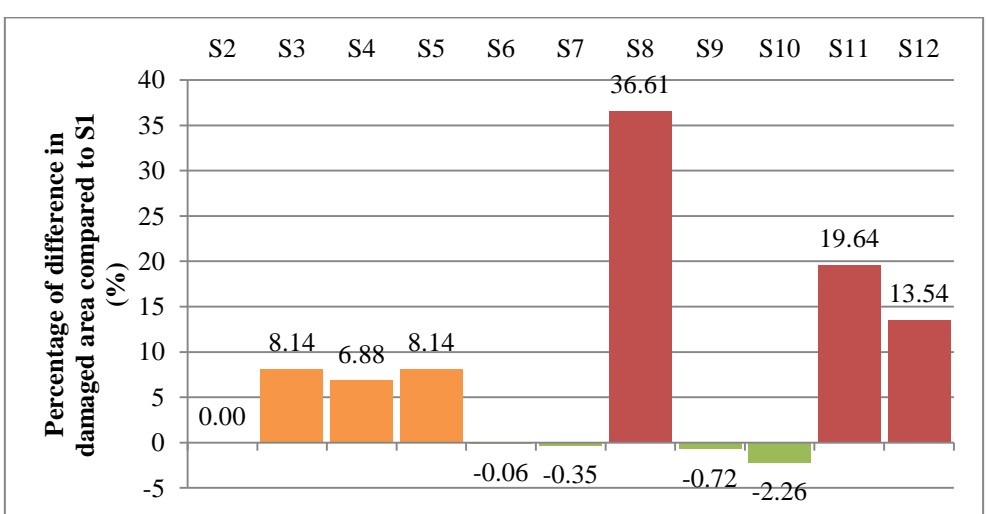

5    **Figure 10: Deviation of total damaged area of all scenarios in relation to S1 (=0)**

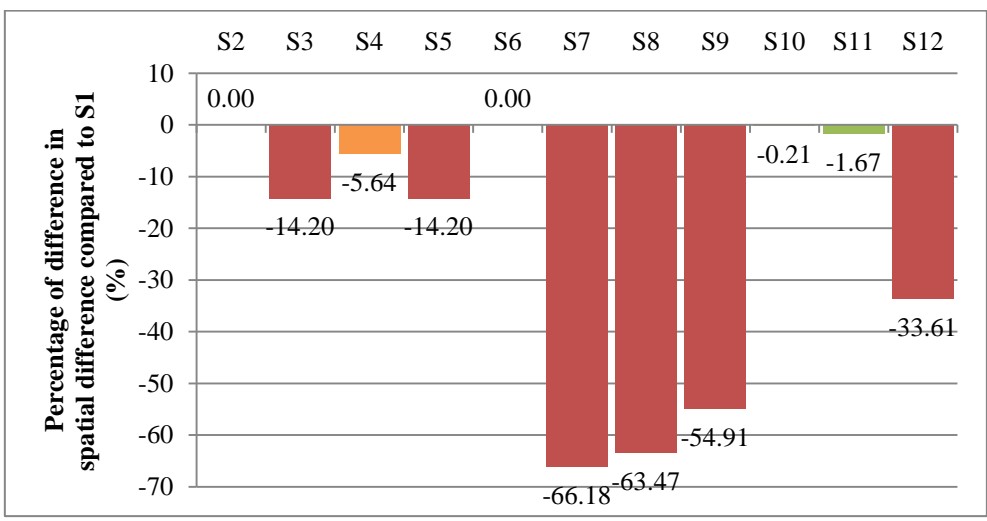

**Figure 11: Deviation of spatial difference of all scenarios in relation to S1 (=0)**

**Table 1: Data used in the Annotto Bay flood risk assessment (Glas et al., 2015)**

| DATA | TYPE | SOURCE |
|---|---|---|
| Landuse | Polygon | NLA (2001) + update based on DigitalGlobe satellite imagery (2010) |
| Roads | Polyline | ODPEM (2013a) |
| Buildings | Point | ODPEM (2013a) |
| Population density | Polygon | Statistical Institute of Jamaica (2012) |
| Average crops values | Table | FAOSTAT (2014) |
| Average building values | Table | ODPEM (2013a) |
| Critical buildings | Point | ODPEM (2013a) |
| 2001 Flood extent | Polygon | ODPEM (2001) |
| Damage functions | Table | Dutta et al. (2003) |

**Table 2: Overview of investigated scenarios in the sensitivity analysis**

| SCENARIO | DESCRIPTION | USED INPUT DATA |
|---|---|---|
| S1 | **Detailed approach** | Land use data<br>Roads (classes) – line<br>2001 flood data<br>Building locations + materials + number of floors |
| S2 | **Building materials and number of floors unknown** | building locations<br>average material values<br>average number of floors |
| S3 | **Building locations, materials and number of floors unknown** | number of buildings known presumed to be equally spread in the urban area |
| S4 | **Building density is calculated based on population density (3 people per building)**<br>Population density is used to determine number of buildings in statistical sectors | |
| S5 | **Building density is calculated based on number of people in study area (3 people per building)**<br>Number of people in the study area is used to determine number of buildings | |
| S6 | **Road classes are unknown**<br>Average values for the width and the cost of the roads are used | |
| S7 | **All roads are unknown and not taken into account**<br>No roads data is used | |
| S8 | **All roads are unknown but taken into account as a percentage of land use**<br>(5% in urban areas, 2% in rural areas)<br>No roads data is used, but the damage is calculated based on a percentage of land use | |
| S9 | **Roads are only used to divide land use polygons – no road damage**<br>Roads are used as a division tool, not to calculate damage | |
| S10 | **Difference between banana plantains and other crops is unknown**<br>In the damage calculations, the same damage factors and maximum costs are used to determine the cost of the crops and the banana plantains | |
| S11 | **Raster approach (10mx10m) based on population density**<br>All input data (vector) is converted to raster data with a resolution of 10 meters | |
| S12 | **Raster approach (30mx30m) based on population density**<br>All input data (vector) is converted to raster data with a resolution of 30 meters | |

**Table 3: Overview of the input data used per scenario**

| | S1 | S2 | S3 | S4 | S5 | S6 | S7 | S8 | S9 | S10 | S11 | S12 |
|---|---|---|---|---|---|---|---|---|---|---|---|---|
| **Building locations** | ✓ | ✓ | | | | ✓ | ✓ | ✓ | ✓ | ✓ | ✓ | ✓ |
| **Number of floors** | ✓ | | | | | ✓ | ✓ | ✓ | ✓ | ✓ | ✓ | ✓ |
| **Building material** | ✓ | | | | | ✓ | ✓ | ✓ | ✓ | ✓ | ✓ | ✓ |
| **Average building values** | ✓ | ✓ | ✓ | ✓ | ✓ | ✓ | ✓ | ✓ | ✓ | ✓ | ✓ | ✓ |
| **Critical buildings** | ✓ | ✓ | | | | | ✓ | ✓ | ✓ | ✓ | ✓ | ✓ |
| **Number of buildings** | | | ✓ | | | | | | | | | |
| **Population density** | | | | ✓ | | | | | | | | |
| **Number of people** | | | | | ✓ | | | | | | | |
| **Roads** | ✓ | ✓ | ✓ | ✓ | ✓ | ✓ | | | ✓ | ✓ | ✓ | ✓ |
| **Road classes** | ✓ | ✓ | ✓ | ✓ | ✓ | | | | | ✓ | ✓ | ✓ |
| **Average road values** | ✓ | ✓ | ✓ | ✓ | ✓ | | | ✓ | | ✓ | ✓ | ✓ |
| **Landuse data** | ✓ | ✓ | ✓ | ✓ | ✓ | ✓ | ✓ | ✓ | ✓ | ✓ | ✓ | ✓ |
| **Banana plants - crops** | ✓ | ✓ | ✓ | ✓ | ✓ | ✓ | ✓ | ✓ | ✓ | | ✓ | ✓ |
| **Average crop values** | ✓ | ✓ | ✓ | ✓ | ✓ | ✓ | ✓ | ✓ | ✓ | ✓ | ✓ | ✓ |
| **2001 Flood extent** | ✓ | ✓ | ✓ | ✓ | ✓ | ✓ | ✓ | ✓ | ✓ | ✓ | ✓ | ✓ |
| **Damage functions** | ✓ | ✓ | ✓ | ✓ | ✓ | ✓ | ✓ | ✓ | ✓ | ✓ | ✓ | ✓ |

**Table 4: Calculated total damage, total damaged area and spatial difference for S1**

| | TOTAL DAMAGE ($) | TOTAL DAMAGED AREA (m²) | SPATIAL DIFFERENCE |
|---|---|---|---|
| **S1** | 7 490 000 | 3 182 000 | 0.048 |

**Table 5: Calculated total damage, total damaged area and spatial difference for S2, S3, S4 and S5 in comparison to S1**

| | TOTAL DAMAGE ($) | | TOTAL DAMAGED AREA (m²) | | SPATIAL DIFFERENCE | |
|---|---|---|---|---|---|---|
| **S1** | 7 490 000 | | 3 182 000 | | 0.048 | |
| **S2** | 8 969 000 | +19.75% | 3 182 000 | +0.00% | 0.048 | +0.00% |
| **S3** | 5 412 000 | −27.74% | 3 441 000 | +8.14% | 0.041 | −14.20% |
| **S4** | 6 997 000 | −6.58% | 3 401 000 | +6.88% | 0.045 | −5.64% |
| **S5** | 5 412 000 | −27.24% | 3 441 000 | +8.14% | 0.041 | −14.20% |

**Table 6: Calculated total damage, total damaged area and spatial difference for S6, S7, S8 and S9 in comparison to S1**

| | TOTAL DAMAGE ($) | | TOTAL DAMAGED AREA (m²) | | SPATIAL DIFFERENCE | |
|---|---|---|---|---|---|---|
| **S1** | 7 490 000 | | 3 182 000 | | 0.048 | |
| **S6** | 7 496 000 | +0.08% | 3 180 000 | −0.06% | 0.048 | +0.00% |
| **S7** | 7 459 000 | −0.41% | 3 171 000 | −0.35% | 0.016 | −66.18% |
| **S8** | 7 490 000 | +0.00% | 4 347 000 | +36.61% | 0.018 | −63.47% |
| **S9** | 7 459 000 | −0.41% | 3 159 000 | −0.72% | 0.022 | −54.91% |

**Table 7: Calculated total damage, total damaged area and spatial difference for S10 in comparison to S1**

| | TOTAL DAMAGE ($) | | TOTAL DAMAGED AREA (m²) | | SPATIAL DIFFERENCE | |
|---|---|---|---|---|---|---|
| **S1** | 7 490 000 | | 3 182 000 | | 0.048 | |
| **S10** | 7 272 000 | −2.91% | 3 110 000 | −2.26% | 0.048 | −0.21% |

**Table 8: Calculated total damage, total damaged area and spatial difference for S11 and S12 in comparison to S1**

| | TOTAL DAMAGE ($) | | TOTAL DAMAGED AREA (m²) | | SPATIAL DIFFERENCE | |
|---|---|---|---|---|---|---|
| **S1** | 7 490 000 | | 3 182 000 | | 0.048 | |
| **S4** | 6 997 000 | −6.58% | 3 401 000 | +6.88% | 0.045 | −5.64% |
| **S11** | 9 692 000 | +29.40% | 3 807 000 | +19.64% | 0.047 | −1.67% |
| **S12** | 8 425 000 | +12.48% | 3 613 000 | +13.54% | 0.032 | −33.61% |