# Peer review of "Analysing the sensitivity of a flood risk assessment model towards its input data"

_Natural Hazards and Earth System Sciences, 2016_

## Referee Comment (RC1) · Anonymous Referee #1 · 24 Sep 2016

The research article: "Analysing the sensitivity of a flood risk assessment model towards its input data" prepared by Hanne Glas, Philippe De Maeyer, Greet Deruyter presents flood damage assessment model that generates a damage map for the region of Annotto Bay, Jamaica. The referee comments are as follows: 1. Does the paper address relevant scientific and/or technical questions within the scope of NHESS? Yes 2. Does the paper present new data and/or novel concepts, ideas, tools, methods or results? Yes 3. Are these up to international standards? Yes 4. Are the scientific methods and assumptions valid and outlined clearly? Yes 5. Are the results sufficient to support the interpretations and the conclusions? Yes 6. Does the author reach substantial conclusions? Yes 7. Is the description of the data used, the methods used, the experiments and calculations made, and the results obtained sufficiently complete and accurate to allow their reproduction by fellow scientists (traceability of results)? I
recommend some corrections. I recommend the section 1.1. Sensitivity analysis includes to chapter 2. Methods. It should be described more preciously how were stated the scenarios for sensitivity analysis. As well described more preciously how were stated total damage costs and total damage area. The authors used average values for the material cost and the building surface area - are these values market values in Jamaica? Or? How was state average maximum road damage? Also how was stated average cost of the crops? How the authors mean the expression in row 29 (chapter 2)... Eleven other scenarios, each with "less or less" detailed input.... Please express clearly (row 29, chapter 5) the resulted indication which data is indispensable and which data can be adopted, replaced or ignored in a risk assessment. Will these data be valid for Jamaica study only or generally? 8. Does the title clearly and unambiguously reflect the contents of the paper? Yes 9. Does the abstract provide a concise, complete and unambiguous summary of the work done and the results obtained? Yes 10. Are the title and the abstract pertinent, and easy to understand to a wide and diversified audience? Yes 11. Are mathematical formulae, symbols, abbreviations and units correctly defined and used? If the formulae, symbols or abbreviations are numerous, are there tables or appendixes listing them? Yes 12. Is the size, quality and readability of each figure adequate to the type and quantity of data presented? Yes. I recommend formal correction - in Fig 4 and Fig. 8 is missing marking S1, S11, S12. Another formal correction is in row 11 in chapter 4 – Figure 9. Formal correction - use jointly: benchmark or bench mark. 13. Does the author give proper credit to previous and/or related work, and does he/she indicate clearly his/her own contribution? Yes, although the newer investigations in the field of interests – flood risk assessment model should be presented in the paper (chapter 1). 14. Are the number and quality of the references appropriate? I recommend including more peer reviewed journals as reference in the Introduction section as well as in the Methodology section. 15. Are the references accessible by fellow scientists? Yes 16. Is the overall presentation well structured, clear and easy to understand by a wide and general audience? Yes, but see comment 7. 17. Is the length of the paper adequate, too long or too short? It is ade-
quate, but see comment 7. 18. Is there any part of the paper (title, abstract, main text, formulae, symbols, figures and their captions, tables, list of references, appendixes) that needs to be clarified, reduced, added, combined, or eliminated? No 19. Is the technical language precise and understandable by fellow scientists? Yes 20. Is the English language of good quality, fluent, simple and easy to read and understand by a wide and diversified audience? Yes 21. Is the amount and quality of supplementary material (if any) appropriate? Yes

---

## Referee Comment (RC2) · Anonymous Referee #2 · 27 Sep 2016

The paper explores the sensitivity of the damage estimation for the inundation caused by the tropical storm "Michelle" in Annotto Bay, Jamaica, on October, 28/29, 2001. For this, a benchmark model with the best available data was defined. Another eleven scenarios with less information, coarser spatial resolution or more rigorous assumptions were then used to investigate differences in the model outcomes. Differences were judged visually and were quantified with regard to the total estimated damage, the damaged area and a metric called spatial difference that quantifies the heterogeneity (and thus detailedness) of a raster map. Three damage types were considered: building, road and crop damage. The authors conclude that the scenario S4 delivers good results – although it is simple – and needs further investigation and that vector data are better than raster data.

While the paper has a reasonable aim, i.e. testing the sensitivity of damage models to

input data, I have a number of major concerns with regard to the set-up of the testing scheme as well as the structure and general presentation of the paper.

Three damage types are considered in the paper: building, road and crop damage, whereof building damage accounts for 90% of the overall losses (see p.7, line 2). First, the choice of these three damage types should be better justified in the paper, ideally on the basis of empirical loss data from Jamaica or other SIDS-countries so that the importance of these three damage types becomes clear and can be discussed later. Second, the sensitivity analysis should not only look at effects on the overall damage estimations, but also at effects on each of the three damage models, separately, in order to have a better understanding of the models' reaction and sensitivity. For this, the damage models used should be explained in more detail and model choices should be better justified. Finally, results should be presented and discussed per damage type and with regard to the initial research question and motivation, particularly the relevance for the analyses for Small Island Developing States (SIDS). For example, the transferability of your assumptions (e.g. 3 persons per household) and the models used (e.g. building damage based on Dutta et al 2003) should be discussed more critically. Actually, the sensitivity of the model to such assumptions should be investigated in the paper. For example, what would be the outcome if you assumed 2 or 4 people per household? A sensitivity analysis should answer such a question.

The benchmark scenario that is based on the inundation of the 2001-event and the best available data to estimate damage should be better justified and descripted. Ideally, it should be accompanied by an event description and official information on its impacts (physical damage and ideally financial losses per damage type or as overall figure). In addition, the use of the best available data as benchmark is somehow contradictory to the findings of Apel et al. (2009), which are mentioned twice as a motivation for this study (p. 2, line 12/13 as well as line 26/27).

Focus and structure of the paper need some improvements, as well. The introduction should summarize the most important findings of the relevant literature as well as the

contribution that this paper (or this case study) adds to the scientific literature. The method section is quite brief, since most of the methods are explained in the results section. You should clearly separate methods, results and discussion. Discussion and conclusion should address the initial research questions as well as the overall motivation of the research to highlight the contribution of this paper to the scientific literature. What can be learned from this analysis – in the specific area, for SIDS countries and beyond?

Conclusions should be based on the findings. The current general conclusion on the suitability of vector and raster data can be questioned in this respect.

Some minor issues: - p.1, line 24/25: Why do you mention flood losses in the UK as example? This does not make sense in the context of this paper. - The crop section (3.4) is not understandable. Provide more basic information on the agriculture in the investigated area and the damage models used. - Present the scenarios and the underlying data and assumption in a matrix table to provide a better overview of the different scenarios - The meaning of the metric "spatial difference" is unclear, in particular with regard to the comparison of different scenarios. - Change model parameters/input data gradually so that the sensitivity of damage models becomes clearer (see above).

---

## Author Response (AR1)

| Comments Anonymous Referee #1 | Response | Changes in manuscript |
|---|---|---|
| I recommend the section 1.1 Sensitivity analysis includes to chapter 2. Methods. It should be described more preciously how were stated the scenarios for sensitivity analysis. As well described more preciously how were stated total damage costs and total damage area. | Thank you for this comment. I will add a few lines to explain the scenarios and the total damage cost and damaged area to improve the clarity of the text. | By changing the structure of the text, it is stated more clearly how the scenarios were constructed and what total damage cost and damage area is. |
| The authors used average values for the material cost and the building surface area – are these values market values in Jamaica? Or? How was state average maximum road damage? Also how was stated average cost of the crops? | The material cost and building surface area are average market values in Jamaica from 2012, that we've received from ODPEM. I will adapt the text to clarify this. The average road damage is based on the average road value in developing countries, as stated by Collier et al (2013). I will add the source for this information. The average crop values were gathered from FAOSTAT. It is true that these average values are not properly explained in the text and I will clarify and adapt this. | I've added the sources of the different damage factors and average values in chapter 3.1 Benchmark model. Furthermore, I've clarified that the damages values used are average market values from either Jamaica or from developing countries. |
| How the authors mean the expression in row 29 (chapter 2)… Eleven other scenarios, each with "less or less" detailed input… | This expression was meant as "less input data or less detailed input data". The second "less" thus belongs to "detailed". | To clarify this, I've added commas in the sentence: "Eleven other scenarios, each with less, or less detailed, input data…". |
| Please express clearly (row 29, chapter 5) the resulted indication which data is indispensable and which data can be adopted, replaced or ignored in a risk assessment. Will these data be valid for Jamaica study | It is true that this section needs some extra clarification. In this research, the scenario that uses population density has the best results. Furthermore, the importance of an adequate road network in order to improve the visual result, has been indicated. In order | A few lines are added to this chapter (chapter 5, last paragraph) in order to clarify the results of the research. |

| | | |
|---|---|---|
| only or generally? | to validate this results, other research areas need to be tested. This will help in determining the possibilities of general use. | |
| Figures – I recommend formal correction – in Fig 4 and Fig 8 is missing marking S1, S11, S12. | Thank you for noticing this, I will add the missing markers in all figures. | Figures are re-entered with the markers in place. |
| Another formal correction is in row 11 in chapter 4 – Figure 9. | This was indeed an error that I've overlooked. Thank you for pointing this out, I will adjust it. | "Fout! Verwijzingsbron niet gevonden" was removed from the text. |
| Formal correction – use jointly: benchmark or bench mark. | I've checked the text and all 'bench mark' notations are corrected into 'benchmark'. | I've checked the text and all 'bench mark' notations are corrected into 'benchmark'. |
| The newer investigations in the field of interests – flood risk assessment model should be presented in the paper (chapter 1) | Thank you for this comment. I will expand the introduction by adding recent developments in the field of flood risk assessment. | In the introduction section, I've added some flood risk assessment tools, as well as some other flood risk assessments done recently. |
| I recommend including more peer reviewed journals as reference in the introduction section as well as in the Methodology section. | By adding the recent developments in the field of risk assessment, more journals will be added. Furthermore, the clarifications needed in the methodology section will also be supported by extra references. | In both sections, more journals were added. |
| **Comments Anonymous Referee #2** | **Response** | **Changes in manuscript** |
| **Major comments** | | |
| Three damage types are considered in the paper: building, road and crop damage, whereof building damage accounts for 90% of the overall losses (see p.7, line 2). First, the choice of these three damage types should be better justified in the paper, ideally on the basis of empirical loss | Thank you for expressing this concern. It is indeed true that the choice of the three types of damages is not justified in the paper. For this research, I've consulted a Multi-Hazard Risk Assessment performed for Annotto Bay by ODPEM, that also discusses the 2001-flood. In this report, the elements at risk were discussed and the three types that | In chapter 2. Methods, I've added a few lines to explain the choice of the three damage types. |

data from Jamaica or other SIDS-countries so that the importance of these three damage types becomes clear and can be discussed later.

Second, the sensitivity analysis should not only look at effects on the overall damage estimations, but also at effects on each of the three damage models, separately, in order to have a better understanding on the models' reaction and sensitivity. For this, the damage models used should be explained in more detail and model choices should be better justified.

Finally, results should be presented and discussed per damage type and with regard to the initial research question and motivation, particularly the relevance for the analyses for Small Island Developing States (SIDS). For example, the transferability of your assumptions (e.g. 3 persons per household) and the models used (e.g. building damage based on Dutta et al 2003) should be discussed more critically. Actually, the sensitivity of the model to such assumptions should be investigated in the paper. For

suffered most damage were buildings, agriculture and roads. (population also suffered, but is not taken into account in this study, since this is a pure economical damage study). I will clarify this in the paper and also add a few references of other studies that use the same types of damage models.

The numbers for the effects on the separate damage models are available as intermediate result of the research, so I can add them to the text. Not all of them are interesting, but it is true that it will help in a better understanding of the sensitivity. In the methods section, I will explain each model more precisely and in the results section, I will clarify the effects on the result per damage type.
Furthermore, I will clarify in the text that '3 persons per household' is not an assumption, but an average for the town of Annotto Bay, gathered from WRA. The damage functions of Dutta et al are chosen since there are many similarities between Jamaica and Japan when it comes to geography and building procedures. I will adapt the text to explain the choice of these functions. I will also add the results in the text of 2 or 4 people per household to show the effects on the overall result and to help explain that some numbers have to be known and cannot be estimated without knowledge of the

The numbers of the separate damage models were added in the results section to better explain the models' sensitivity.
In chapter 3.2, I've adapted the text to clarify that 3 people per household is not a presumption, but an average, gathered from WRA.
In chapter 3.1, I've added the reasons of choosing the damage functions from Dutta et al., for building damage as well as for crop damage.
I've added the resulting damage costs of the model run with an average of 2 or 4 people per household in the discussion section. In the methodology section, I've added an extra line to

| | | |
|---|---|---|
| example, what would be the outcome if you assumed 2 or 4 people per household? A sensitivity analysis should answer such a question. | region. | emphasize the importance of an accurate number of people per household. |
| The benchmark scenario that is based on the inundation of the 2001-event and the best available data to estimate damage should be better justified and descripted. Ideally, it should be accompanied by an event description and official information on its impacts (physical damage and ideally financial losses per damage type as overall figure). In addition, the use of the best available data as benchmark is somehow contradictory to the findings of Apel et al. (2009), which are mentioned twice as a motivation for this study (p. 2, line 12/13 as well as line 26/27). | I will adapt the paper and add the numbers that we have on the actual event to help justify the choice of benchmark. However, not all information is available for the real event, so a complete justification cannot be added.

I understand that you see it as contradictory to use the best available data with the findings of Apel et al, mentioned in the introduction. This research, however, is not a search to lower uncertainty of the output model, but a test to see which data has the highest influence on the result of the model, to test its sensitivity. Therefore, we chose to work with the best available data, as done before in many other studies, to then check if all input data is necessary to generate the same result. I agree that the research goal should be stated more clearly and I will adapt the text to clarify this. Of course, this does not mean that uncertainty is not important and in further research, this will be investigated. | The numbers available from the 2001 flood are added at the end of chapter 3.1 Benchmark map. Furthermore, the research aim is explained more clearly in this section, explaining the choice of S1 as benchmark result. |
| Focus and structure of the paper need some improvements, as well. The introduction should summarize the most important findings of the | I will add some recent papers on flood risk assessment and the most important findings in the introduction section. | The papers were added to the introduction to help state the importance of the research. |

| | | |
|---|---|---|
| relevant literature as well as the contributions that this paper (or this case study) adds to the scientific literature.

The method section is quite brief, since most of the methods are explained in the results section. You should clearly separate methods, results and discussion. | Thank you for your view on the methods and results section. It's true that these are not clearly separated. This is due to the fact that the paper discusses 12 scenarios, each with their own methodology. Explaining these all in the methods section before showing any results, seems confusing for the reader. That is why the authors propose to change the structure of the text as follows: chapters 2 and 3 would be combined in one chapter, named 'Methods and Results'. Then, each type of damage would be discussed separately, first the methodology, than the results: | The structure of the text was adapted as suggested in order to clearly separate the methods from the results. |

2.   Methods and Results
    2.1 Benchmark map
       2.1.1    Method
       2.1.2    Result
    2.2 Building damage sensitivity
       2.2.1    Methods
       2.2.2    Results
    2.3 Road damage sensitivity
       2.3.1    Methods
       2.3.2    Results
    2.4 Crops damage sensitivity
       2.4.1    Methods
       2.4.2    Results
    2.5 Data type sensitivity
       2.5.1    Methods
       2.5.2    Results

I know it is not the standard way of structuring a paper, but considering the content of the paper, this structure gives a clear overview of the research. Does this seem like a good possibility for you?

I will rewrite the discussion and conclusions

| | according to your comments. | |
|---|---|---|
| Discussion and conclusion should address the initial research questions as well as the overall motivation of the research to highlight the contribution of this paper to the scientific literature. What can be learned from this analysis – in the specific area, for SIDS countries and beyond? | | The discussion is rewritten to clarify the results of the study, while the conclusion is rewritten to emphasize the answers given on the research questions and the important findings. |
| Conclusions should be based on the findings. The current general conclusion on the suitability of vector and raster data can be questioned in this respect. | That is true. I will adjust this conclusion. The raster data scenarios had less accurate results. This can be due to resolution and generalization of the vector data. I will adapt the text that this conclusion clearly reflects the findings of this study. | The paragraph of the conclusion concerning the raster data was adapted and now reflects the results of the research. |
| **Minor comments** | | |
| P1., line 24/25: Why do you mention flood losses in the UK as example? This does not make sense in the context of the paper. | This is definitely a fair point since the context of the paper focuses on flood losses in developing countries. Since this does not contribute to the paper, I opt to remove this example from the text. | The UK example has been removed from the paper. |
| The crop section (3.4) is not understandable. Provide more basic information on the agriculture in the investigated area and the damage models used. | I will add information on banana plants and on other crops, frequently grown in Jamaica. I will explain how the plants cope with water and how the damage functions are generated. This will help in clarifying the | Extra information on banana plants and other crops was added in chapter 3.1. The crop section (3.4) was rewritten to make it |

| | overall crop damage model. | understandable. |
|---|---|---|
| Present the scenarios and the underlying data and assumption in a matrix table to provide a better overview of the different scenarios. | Thank you for this idea, it will help in clarifying the differences in scenarios. I will add this matrix to the general methodology. | I've added the matrix as Table 3, providing an overview of what data is used in which scenario. |
| The meaning of the metric "spatial difference" is unclear, in particular with regard to the comparison of different scenarios. | I understand the confusion since the spatial difference is calculated as a percentage. In the comparison with other scenarios, another percentage (the difference with S1) is then calculated. To avoid this confusion, the spatial difference will be calculated as an absolute number, and a formula with the exact calculation will be added to the text. Furthermore the percentage of difference with S1 will be added to the tables with the results of other scenarios, so the reader can immediately get an idea of the similarities between scenarios. This will not only be done for the spatial difference, but also for the total damaged area and the total damage cost. | The equation for spatial difference is added in Eq. (1) in chapter 2. |
| Change model parameters/input data gradually so that the sensitivity of damage models becomes clearer (see above). | The parameters are not chosen randomly, but are seen as a form of input. Since this research aims to test the sensitivity of the model towards different types of input data, there was opted not to change the parameters gradually, but to use different types of input data and to change the level of detail of the available data. This is, however, a very interesting point of view in regards to further research and validating results of different study areas. | The text was not adapted but this comment is definitely an interesting point of view for further research. |

[revised manuscript text omitted]